# Mitochondrial Functions, Cognition, and the Evolution of Intelligence: Reply to Commentaries and Moving Forward

**DOI:** 10.3390/jintelligence8040042

**Published:** 2020-12-08

**Authors:** David C. Geary

**Affiliations:** Department of Psychological Sciences, Interdisciplinary Neuroscience, University of Missouri, Columbia, MO 65211-2500, USA; GearyD@Missouri.edu; Tel.: +1-573-882-6268; Fax: +1-573-882-7710

**Keywords:** mitochondria, cognition, intelligence, health, evolution

## Abstract

In response to commentaries, I address questions regarding the proposal that general intelligence (*g*) is a manifestation of the functioning of intramodular and intermodular brain networks undergirded by the efficiency of mitochondrial functioning (Geary 2018). The core issues include the relative contribution of mitochondrial functioning to individual differences in *g*; studies that can be used to test associated hypotheses; and, the adaptive function of intelligence from an evolutionary perspective. I attempt to address these and related issues, as well as note areas in which other issues remain to be addressed.

## 1. Introduction

I thank the editors and commenters for thoughtful critiques and questions regarding the proposal that mitochondrial functioning is the most fundamental biological process contributing to human cognition and links cognition to health and aging ([47], [48], [49]). I cannot address all of the critiques and questions in detail and will focus on the most central of them: specifically, the relative contributions of mitochondria to individual differences in *g* ([30]; [78]; [99]; [112]; [119]); empirical approaches to testing the hypothesis ([13]; [78]; [113]); and, the adaptive function of intelligence from an evolutionary perspective ([27]; [113]). Many of the more specific critiques and questions are addressed in the context of these broader issues and summary responses to them are provided in Table 1.

To begin, [30] ([30]) notes that mitochondria are not the only mechanism associated with energy production and there are multiple pathways involved in the generation (whether or not it is in the mitochondria) of adenosine triphosphate (ATP), including transport of substrates needed for this production. He rightly argues that the focus should be on the entire system involved in ATP production, of which mitochondria are only one part. I completely agree and noted this in the original proposal: “I have situated cellular energy production and functioning, *largely* supported by mitochondria, at the most basic level’ ([47]). I also noted that “[variation] in energy availability can result from differences in the substrates (e.g., pyruvate) available to fuel the process” ([47]), but agree that my focus on mitochondria may have given the impression that the system is less complex than it actually is. Nonetheless, the mitochondria produce most but not all—glycolysis is an additional source of cellular energy ([88])—of the energy needed for brain development and functioning and are thus a critical mechanism, although I fully agree that this is only part of a more complex system of energy dynamics.

Perhaps the most central question to emerge in the commentaries is the testability of the model and a correlated issue of whether mitochondrial functioning—broadly meaning ATP production, influences on the availability of associated substrates, control of oxidative stress and other processes—is influenced by environmental factors. As shown in Figure 1, mitochondrial functioning is very sensitive to a variety of social and environmental factors, including chronic psychosocial stress ([91]), glucose homeostasis as related to diet and activity levels ([89]), and toxin exposure ([15]), among others. In Section 3, I illustrate this sensitivity with discussion of how glucose homeostasis (e.g., as related to diet) contributes to mitochondrial health and cognitive functioning, and in doing so illustrate one way in which the theory might be tested ([13]; [99]). Before turning to this literature, I consider issues regarding the relative importance of mitochondrial functions in the context of individual differences in *g*.

## 2. Relative Importance of Mitochondria

The key issues addressed in the first subsection concern the relative influence of mitochondrial functions on estimates of individual differences in *g*, broader cognitive abilities, and environment influences, including the Flynn effect ([13]; [78]; [99]; [112]; [113]). The key issue addressed in the second subsection concerns the relation between mitochondrial genes and individual differences in *g* ([101]; [119]).

### 2.1. Cognition and Mitochondria

#### 2.1.1. Mitochondrial Contributions as a Proportion of *g*

At this time, the relative contributions of individual differences in mitochondrial health to individual differences cognitive functioning cannot be determined. The upper limit will be the proportion of variation in cognitive abilities that is captured by estimates of *g*. [112] ([112]) argues that this upper limit could be lower, perhaps 35% or even lower still, than the 50% I suggested ([47]). The variability in the estimates for *g* are related in part to differences in the specific measures included in the test batteries used in one study or another ([17]; [60]).

As described in the *Aging and g* section of [47] ([47]), stronger correlations and thus a higher estimate of variance explained by *g* is anticipated for batteries that include more complex tasks. This is because mitochondrial energy production is hypothesized to provide a ceiling on optimal performance and the ceiling is more likely to be reached by more people (revealing greater individual differences) for tasks that require the sustained use of distributed (intermodular) brain systems, not simple processes (e.g., reaction time) as suggested by [78] ([78]) and [101] ([101]); the former could reflect variation in the ability to maintain attentional control at the behavioral level, as suggested by [13] ([13]).

Moreover, the variance explained by a *g* factor should also depend on the diversity of the sample. Young, healthy and well-educated samples, as in the [87] ([87]) study mentioned by [112] ([112]), will likely produce lower estimates than will more representative samples. By analogy, individual differences in cardiovascular fitness will be more apparent during a stress test than during a casual walk, and variation in fitness will be more apparent in the general population than in young, college-educated adults. In any event, a recent analysis revealed that the positive manifold is universal and explains approximately 46% of the covariance among cognitive measures ([124]), as I originally suggested ([47]).

Even if we accept [112]’s ([112]) lower estimate for *g*, there is still ample room for substantive contributions of mitochondrial health to cognitive functioning but estimating the extent of these contributions will require the types of interdisciplinary studies described in Section 3.

#### 2.1.2. Mitochondria and Alternative Models of Human Cognition

[112] ([112]) argues that I chose to ignore the Cattell–Horn–Carroll (CHC) model and related theories in favor of *g*, but this is not the case (see also [78]). Broad factors other than *g* are clearly important for understanding human cognition. As was noted in Table 1 of the original article ([47]), if the positive manifold is due to overlapping processes across different cognitive measures—an alternative model to *g*—and thus a statistical artifact, then this would falsify my model. Moreover, if my proposal regarding the structure of *g* is correct then models such as CHC become the central focus of the organization of human abilities (see [48]). This is because *g* will not manifest as a psychologically measurable ability, but rather as individual differences in abilities and intraindividual developmental (e.g., with normal aging) and disease-related (below) changes in abilities.

From this perspective, *g* might be considered a composite measure of the functioning of the brain networks that contribute to cognition (in agreement with [99]), systems whose functioning is ungirded by more basic mechanisms, including mitochondria and other factors that influence energy availability ([30]). Within-person longitudinal changes in estimates of *g*, as with normal age-related changes in adulthood, might then provide a global estimate of the rate of degradation of these systems from their peak. As noted in the original proposal ([47]), the most sensitive tests of this decline are predicted to be those that engage the largest distributed intermodular networks, such as those associated with fluid ability ([62]). This is because these systems are very energy intensive ([12]) and are therefore the most sensitive to declines in energy production. At the same time, deficits in these domains would be found in the context of broad declines across cognitive abilities, as appears to be the case ([116]).

The basic prediction is that the vulnerability of the cognitive system will be directly related to the amount of neural tissue needed to support the associated ability and the amount of time needed to complete any associated tasks. [101] ([101]) note that short-term memory (STM) is embedded in working memory (WM) tasks and that the latter involve the additional processes of information manipulation ([23]). Given that WM by definition engages a more complex system of cognitive processes, my prediction is that performance on these measures will decline more sharply than will performance on STM measures, if there are broad declines in mitochondrial health (e.g., as with normal aging).

#### 2.1.3. Mitochondria, Environmental Conditions and the Flynn Effect

From an individual differences perspective, *g* could be considered a composite estimate of the genetic and environmental influences on the development and current functioning of the brain systems that support cognition. [101] ([101]) and [113] ([113]) broach the associated issue of whether this perspective is useful for understanding individual differences in cognition among healthy adults. The answer will have to await the development of more sensitive measures of individual variation in mitochondrial functions, and these are being currently developed (see Section 3.1).

Until that time, I note that the young and healthy samples described by [101] ([101]) are not representative of the condition of most people during our evolutionary history or even many people in developing nations today (for reviews see [45]; [115]). The abundance of food, reduction in infections, improvements in living conditions and so forth are evolutionarily novel and likely mask the effects of individual differences in mitochondrial functions in the modern world, or at least do so for a larger segment of the population and the lifespan than in our ancestors.

On this view, broad improvements in public health, including better nutrition, reductions in disease and toxin exposure, and reductions in chronic social stressors will help to maintain mitochondrial health that in turn enable a fuller expression and maintenance of genetic potential ([98]). The result is an improvement in cognitive abilities that essentially reflects a rescue of genetic potential by reduction in environmental risk factors, such as poor nutrition ([94]). This is in keeping with [101]’s ([101]) interpretation of [96]’s ([96]) finding that creatine supplementation—which increases the efficiency of mitochondrial energy production (among other things)—increased the cognitive performance of young healthy vegetarians who typically have below-average creatine levels, but similar treatments are not as helpful for healthy adults without creatine deficits ([6]).

Improvements in nutrition and living conditions more broadly could have contributed to the well-known secular increase in performance on intelligence and cognitive tests in developed nations, that is, the Flynn effect ([41]; [32]; [37]; [92]; [98]). If so, we would expect larger Flynn-effect gains on measures of fluid than crystallized intelligence, and this seems to be the case. The approach could also explain at least some of the differences in average cognitive abilities across people living in developed and developing countries.

Before turning to the next section, [78] ([78]) raised a related question, that is, if mitochondrial functioning is critical to *g*, then should not people high in *g* also have advantages in other domains, such as running speed? The answer is: No. Evolutionary selection can independently act on traits above the level of mitochondria. There are different advantages to traits such as running speed and intelligence and thus we would expect them to be more or less elaborated in different contexts. Mitochondrial health will influence the extent to which these traits are fully expressed within each individual, but this is not the same as high interindividual correlations between these traits. Nonetheless, a common link might be found with decrements in mitochondrial functioning, whereby *intraindividual* changes in performance across traits are correlated (e.g., with normal aging).

### 2.2. Genes and Mitochondria

#### 2.2.1. Genome-Wide Association Studies, Mitochondrial Proteins, and *g*

[119] ([119]) argue that genome-wide association studies (GWAS) that have identified single-nucleotide polymorphisms (SNPs) associated with individual differences in cognition are incompatible with the mitochondria model ([47]). One reason is that many of the identified SNPs are typically associated with various aspects of brain and neuronal functioning (e.g., [22]; [26]). These types of studies use SNPs to predict cognitive or educational outcomes and thus might be biased to more readily detect SNPs associated with biological processes that are more directly related to these outcomes than are mitochondrial functions.

A complimentary and bottom-up approach would be to examine the relation between mitochondrial gene-product proteins and protein networks and individual differences in cognitive functions or intraindividual differences in the stability of cognitive functions with aging. Due to technical challenges (e.g., needing brain issue), these types of studies are not nearly as common as the standard GWAS studies noted by [119] ([119]). Nevertheless, they have the potential to test the mitochondrial model and to broaden the search for the biological foundations of cognitive ability more generally.

[127] ([127]) conducted a protein-wide association study of longitudinal change (6 months to 20 years) in cognitive ability among older adults; proteins were analyzed from brain tissue after the participants had died. They found that 579 proteins and some interactions among them were associated with cognitive stability with aging. Stability was associated with low levels of expression for some proteins, such as those associated with chronic inflammation (see Section 3.2.2), and high levels of expression for other proteins. The most abundant of the latter included proteins associated with mitochondrial functions, neurons, and synaptic transmission; they note that “mitochondrial proteins were featured prominently among cognitive trajectory related proteins … the 350 proteins with increased abundance in cognitive stability participate in mitochondrial activities and synaptic transmission in neurons” (p. 4). With control of pathological processes (ß-amyloid plagues), protein networks associated with synaptic and mitochondrial functions remained significantly related to cognitive stability.

The results are consistent with the GWAS studies listed by [119] ([119]) and indicate that their dismissal of mitochondrial functions is premature. Indeed, [127] ([127]) point out that core synaptic and neural functions are dependent on mitochondrial health, whereby “157 proteins of the 579 cognitive trajectory-associated proteins are mitochondrial proteins and of these, 122 are located in the mitochondria in either the pre- or postsynaptic density” (p. 9).

[119] ([119]) also noted that many of the SNPs identified in GWAS studies are from intergenic regions with no known function and have been identified due to linkage disequilibrium, that is, correlated transmission with functional genes. [52] ([52]) sought to address this issue by focusing on GWAS-identified SNPs correlated with intelligence and with known biological functions. Using Bayesian and other techniques, they identified the SNPs and associated genes most likely to be causally related to individual differences in intelligence. The most plausible SNP coded for an enzyme that is critical for the functioning of the mitochondrial electron transport chain associated with ATP production, that is, energy production expressed throughout the body. Other identified genes were associated with pyramidal cells and synapse formation; mitochondrial membrane potential; neural stem cells; and, cell growth and differentiation.

#### 2.2.2. Parental Genetic Influences on Mitochondrial Functions

A more general question is whether the parent-child correlations for intelligence should be higher for mothers than for fathers, given that mtDNA are inherited solely from mothers ([78]; [101]). There is no straightforward reason to believe there will be stronger mother-child than father-child correlations for intelligence, because the mtDNA include just 37 genes, only 13 of which are directly related to the electron transport chain that produces ATP. This is in comparison to more than 1000 biparental nuclear DNA (nDNA) genes that support mitochondrial energy production and other mitochondrial functions ([16]): As noted in the original article ([47]), the nDNA genes that support mitochondrial functions evolved as a compensatory mechanism for degradation of the mitochondrial genome ([54]).

My point in the original article was that the uniparental inheritance of mtDNA could contribute to the greater *variation* in intelligence among males than females ([61]), and especially contribute to the over-representation of males at the lower end of the distribution. This is because uniparental inheritance means that mutations that disadvantage males but are neutral or beneficial to females cannot not be eliminated by natural selection ([9]). These mutations would presumably result in more deficits in brain and cognition for males than females, but the evolution of compensatory nDNA and the potential importance of mtDNA–nDNA interactions as related to mitochondrial functions do not result in a straightforward relation between maternal intelligence and offspring intelligence.

Moreover, mt mutations that differentially compromise male cognition will not necessarily be expressed in their mothers’ intelligence, because mutations that compromise the latter will eventually be eliminated by evolutionary selection. The associated process of purifying selection (for females) would disrupt any mtDNA-related correlations between mothers’ and sons’ intelligence. Rather, these correlations would be driven by nDNA and as such should be similar to father-child correlations, as appears to be the case ([125]). Note that this does not mean that maternally inherited mtDNA mutations cannot affect boys’ cognition more strongly than that of girls. Rather, any such mutations will not necessarily be detected with simple mother-son phenotypic correlations.

## 3. Empirical Studies and Testing the Hypothesis

As noted in several commentaries ([13]; [78]; [99]; [113]; [112]), the critical next steps include empirical tests of predictions derived from the model. The key hypothesis is that mitochondrial health is linked to cognitive performance. One specific prediction is that biomarkers that are indicative of good mitochondrial functioning will predict better cognitive performance, especially on complex tasks. A second specific prediction is that the progression of diseases that compromise mitochondrial functioning will be associated with parallel declines in cognitive functioning. Finally, interventions that promote mitochondrial health should result in improvement in cognitive performance. Overviews of research related to each of these specific predictions is provided below.

### 3.1. Mitochondrial Biomarkers

Although much remains to be determined, efforts are underway to develop biomarkers of mitochondrial health and dysfunction. In the near future, these biomarkers might prove to be useful in the study of individual differences in cognitive functioning and longitudinal change in this functioning ([90]). One such marker is mtDNA copy number. The latter is an estimate of the number of functioning mitochondria per cell, which generally declines with aging and the progression of various diseases ([81]). There are currently several methods available for the assessment of copy number and others in development ([72]); other potential markers include indicators of systemic inflammation (below) and the proportion of wild-type (original) and mutated mtDNA.

There are several preliminary studies that suggest this might be a useful approach ([80]; [105]), with the caution that the relation between mtDNA copy number measured in peripheral cells (e.g., white blood cells) in many of these studies and mtDNA copy number in brain cells is not well understood. In the larger of these studies (*N* = 1067), Mengel-From et al. found steady mtDNA copy numbers through age 48 years in healthy individuals and age-related declines thereafter. For older individuals, higher mtDNA copy numbers were associated with better physical health and higher cognitive performance; “higher mtDNA copy number was consistently associated with higher cognitive composite score and MMSE [Mini Mental State Examination]” ([80]). In a sample of older women, J. W. [69] ([69]) found that higher mtDNA copy number was associated with higher MMSE scores (*r* = 0.33), controlling age and years of education, and [105] ([105]) found that higher copy number was associated with better list learning and memory scores in older adults (*r* = 0.46).

In a unique study noted by [78] ([78]), [11] ([11]) examined the relation between placental mtDNA copy number taken at the time of birth and intelligence 8 to 15 years later. A doubling of copy number was associated with a 2-point gain in IQ. Among monozygotic twins who shared the same placenta, the twin with higher mtDNA levels (in cord blood) had higher IQ scores than the cotwin (*r* = 0.41). The potential mechanism is prenatal competition between the twins for maternal resources; the twin receiving more resources potentially having higher mtDNA copy numbers in cord blood that in turn may have influenced prenatal brain development. Matzel et al. suggested that this was a prenatal environmental effect and not related to specific advantageous maternal mtDNA. I agree. The point is that, whatever the source, additional mitochondria might provide benefits to prenatal brain development. [The related issues regarding the contributions of mtDNA and nDNA to mitochondrial functioning are addressed in the section *Uniparental Inheritance of Mitochondria and Variation in g* (see [47]).]

At the same time, there are tissue-specific variations in mtDNA copy number and thus the best noninvasive method (e.g., in blood cells) for their measurement in the central nervous system is not yet known ([123]). Moreover, there are some disease conditions that can result in a compensatory increase in mtDNA copy number, possibly with a corresponding increase in the number of mtDNA mutations, and thus a linear relation between copy number and cognitive functions might not always emerge ([58]). Despite these caveats, the development of biomarkers of mitochondrial health will provide unique opportunities for future interdisciplinary work involving cognitive and biomedical scientists.

### 3.2. Mitochondrial-Related Disorders and Cognition

There are progressive diseases, including mitochondrial disorders, that compromise mitochondrial functions and are associated with cognitive declines. Prospective studies of cognitive changes associated with disease progression provide an indirect test of the hypothesized relation between mitochondrial health and cognitive function. Moreover, the pattern of cognitive decline should be similar to that found for normal age-related cognitive changes and perhaps the Flynn effect, including larger changes for more attention-demanding tasks (e.g., fluid intelligence) relative to less difficult tasks, such as short-term memory (e.g., forward digit span), memory retrieval (e.g., vocabulary), or simple reaction time.

The parallel between cognitive declines with disease progression and normal age-related changes follows from the hypothesis that compromises to mitochondrial functioning are a common underlying and causal mechanism, and inconsistent with [78]’s ([78]) argument that the relation between cognition and health is simply due to health-related decisions and behaviors (see also [29]). At this time, only indirect evidence can be marshalled to evaluate the hypothesis. The study of mitochondrial disorders is one approach, although it is not as straightforward as it might seem. The study of cognitive changes associated with obesity-related compromises in mitochondrial health is another such disease ([89])

#### 3.2.1. Mitochondrial Disorders

On the basis of my proposal ([47], [48]), mitochondrial disorders should be associated with lower performance across cognitive domains, as noted by [78] ([78]) and [112] ([112]). As an example, Stankov cites one study in which some but not all individuals with a mitochondrial disorder showed cognitive impairments. These types of studies are not necessarily evidence against my proposal, because there are different forms of mitochondrial disorder with different degrees of severity and different rates of disease progression ([20]; [102]).

At this point, a well-designed study of the cognitive competencies of individuals with mitochondrial disorders and changes in these competencies as the disease progresses remains to be conducted. A recent review of extant studies shows that cognitive deficits are more common on complex attention-demanding measures (e.g., executive functions) than on measures of crystallized intelligence; that asymptomatic individuals often show only subtle deficits but deficits are common in symptomatic individuals (e.g., showing vision difficulties, seizures); and, cognitive abilities decline as the disease progresses ([82]; [83]; see also [39]).

#### 3.2.2. Obesity, Diabetes, and Inflammation

The availability of energy substrates (lipids, glucose), including chronic over availability resulting in obesity and physical exercise that consumes these substrates (below), can influence long-term mitochondrial health and functioning ([89]). There are many details that cannot be adumbrated here but the key is that mitochondria are highly responsive to metabolic state, and chronic obesity-related disruption of glucose homeostasis is associated with cognitive declines and accelerated aging that are at least in part related to compromised mitochondrial functions.

An overabundance of calories is associated with increased risk of insulin resistance, Type II diabetes, atherosclerosis, and other health issues. Among other things, swollen adipose tissue leaks lipids (fatty acids) that can trigger an immune response and chronic inflammation ([110]). The oversupply of energy also overwhelms mitochondrial energy production systems that in turn exacerbate inflammation and increase cell and DNA damaging oxidative stress ([28]; [111]). The mechanisms that would typically keep this damage in check are simultaneously compromised, resulting in further damage ([89]).

The associated inflammatory molecules can disrupt the blood brain barrier and their movement into the central nervous system can result in neuroinflammation ([95]). One consequence is a reduction in the energy producing capacity of mitochondria, among other things (e.g., disruption of intracellular signalling; [10]). There are defensive mechanisms to maintain energy supplies—for instance increased generation of new mitochondria but often with the cost of increased levels of mtDNA mutations—but declines continue (without intervention) and eventually impact system functioning. [89] ([89]) estimated that these changes, if chronic, are equivalent to 5 to 20 years of normal aging moving from middle to old age (see also [114]).

As with mitochondrial disorders, the definitive study of obesity-related disruptions in mitochondrial functioning as related to cognition remains to be conducted. There is nevertheless good reason to suspect a substantive relation. There are moderate but consistent (Effect Size, ES ≈ 0.3 to 0.45) differences in various components of executive functions comparing obese to normal weight individuals and even moderate deficits in inhibition and working memory comparing overweight to normal weight individuals (ES ≈ 0.1 to 0.2; [129]). These patterns are likely bidirectional, with poor executive functions associated with difficulties in regulating food intake, as well as compromises in executive functions resulting from the just described biological mechanisms ([104]; [110]).

In a sample of healthy adults (*N* = 180), [109] ([109]) found lower fluid intelligence in obese as compared to normal weight individuals. A series of path models suggested that the differences were mediated by inflammation, one indicator of compromised mitochondrial functioning; an alternative model in which fluid intelligence caused weight differences did not fit the data as well as the inflammation model. Stronger evidence is found with prospective studies of the relation between baseline risk factors, such as chronic inflammation, and rate of cognitive decline and impairment with aging ([19]; [71]; [100]; [128]). As an example, a 10 year longitudinal study of 45- to 69 year olds revealed accelerated declines in reasoning and memory abilities for individuals with indications of chronic inflammation at baseline ([106]). The declines were equivalent to an additional 1.7 years of normal age-related cognitive changes over the 10 years of the study.

Studies using Positron Emission Tomography (PET) indicate declines in glucose utilization with the progression of metabolic diseases (e.g., diabetes). [121] ([121]) found that obesity was associated with lower baseline levels of glucose utilization among young, healthy adults (*M* = 34 years) that in turn was associated with lower performance on a battery of neuropsychological tests. They suggested that the lower baseline glucose utilization was due to reduced grey matter volume, but this remains to be determined.

In a large study of older adults (*M* = 61 years), [126] ([126]) found that insulin resistance was associated with lower overall (*β* = −0.29) and regional differences in brain glucose metabolism. The strongest changes were in the hippocampus and medial temporal lobe, which has a high density of insulin receptors and high energy demands because of continual neurogenesis. Higher glucose metabolism (an indication of mitochondrial functioning) in these regions was associated with better immediate memory (*β* = 0.32) and verbal learning (*β* = 0.31) but had weaker relations to working memory (*β* = 0.12) and processing speed (*β* = 0.20).

With disease progression these deficits become widespread and are found at multiple levels, including reductions in grey and white matter, reduced hippocampal neurogenesis, and reduced mitochondrial trafficking within axons that in turn disrupts synaptic functions, among other things ([86]; [33]). Studies of older at-risk individuals (e.g., diabetes) indicate declines in glucose metabolism throughout the prefrontal cortex and parietal regions ([8]; [67]), that is, regions associated with working memory and fluid intelligence ([62]).

A recent meta-analysis suggested that for otherwise healthy people, many of these declines may not be detectable until years after the onset of inflammation or other aspects of disrupted glucose homeostasis ([5]; [19]), although high fat diets might result in changes fairly quickly ([34]). As cognitive and functional abilities begin to decline, there is often an increase in brain glucose metabolism that likely reflects use of compensatory cognitive strategies and biological mechanisms (e.g., increased mitochondrial biogenesis; [59]). The compensatory mechanisms temporarily maintain functions but could accelerate mitochondrial and neural damage and eventually result in precipitous declines in cognition and day-to-day functioning.

From the perspective of my model, deficits associated with disrupted glucose homeostasis (e.g., as indexed by measures of systemic inflammation) should result in a pattern of cognitive decline that varies with the energy demands of the cognitive task—these could be indexed by PET studies or the size and complexity of the supporting brain systems as identified in *f*MRI studies (see critique by [119])—and track the pattern of age-related cognitive declines. This would include shaper declines in fluid intelligence, measures of active learning, and attentional control and later and more shallow declines in crystallized abilities, such as vocabulary ([47]). To be sure, there are many pathological changes associated with chronic disruptions in glucose homeostasis, above and beyond declines in mitochondrial functioning, but declining mitochondrial health is an important component of the disease process ([89]), and the study of the pattern of cognitive declines associated with this progression provides a means to assess the plausibility of my model ([47], [48]).

An important corollary noted by [78] ([78]) is that the relation between mitochondrial declines and cognitive declines is not linear, because of compensatory mechanisms that maintain energy production. As noted, the maintenance of energy production may be associated with increased mitochondrial numbers but with increases in the proportion of mutated mtDNA (vs. wild type). If so, then the proportion of mutated to wild-type mtDNA copy numbers might eventually prove to be a useful marker of impending cognitive declines.

### 3.3. Mitochondrial Health and Cognition

If disruption of glucose homeostasis and associated issues with inflammation and mitochondrial dysfunction is causally related to the just-described cognitive declines, then interventions that address the disruption should improve cognitive functioning or reduce the rate age-related decline. These types of interventions exist and range from substantive weight loss (e.g., involving bariatric surgery), insulin administration, use of drugs to reduce oxidative stress, and exercise, among others ([10]; [103]; [108]). There is evidence that many of these interventions can improve cognitive functions, although individuals differ in their responsiveness to such interventions for reasons that are not yet fully understood.

I focus here on weight loss, exercise, and their combination, because studies of nonhuman animals and humans have provided key insights into the biological changes associated with such interventions and correlated changes in cognition. Overall, a combination of calorie restriction, weight loss, and exercise can reverse many of the above-described effects associated with too much energy and disrupted glucose homeostasis. These include the upregulation of a variety of protective mechanisms, including upregulation of mechanisms that support mitochondrial energy production, DNA repair and antioxidant production, as well as removal of damaged cells and damaged mitochondria within cells. The result is improvements in functioning across intracellular to intermodular systems ([77]).

In a large-scale randomized controlled trail (RCT), [85] ([85]) followed nearly 1200 older individuals (66 to 77 years old) at-risk for dementia for 24 months; these individuals were average or lower in overall cognitive ability. The intervention included diet, exercise, and cognitive stimulation. The combination resulted in modest (ES = 0.13) cognitive benefits overall but more substantive benefits for executive functions, processing speed, and delayed memory recall. The results are similar to those found in other RCTs with healthier adults, where ESs are approximately 0.15 ([107]). A more recent meta-analysis included estimates of longitudinal change in cognition following weight loss and RCTs for very obese individuals, and found more substantive benefits ([120]). For the longitudinal studies, the ESs for various attention, executive functions, and memory measures ranged from 0.30 to 0.66, with no improvement on language measures (language was not frequently assessed in these studies and thus the ES is unreliable). The repeated assessment in such studies could, however, inflate the ES estimates, but the RCTs also showed improved cognitive performance, especially with the combination of diet and exercise; attention (ES = 0.44), memory (ES = 0.35), language (ES = 0.21), with no effect for executive functions but this was assessed in only two studies. The authors concluded that the gains might be related to reductions in insulin resistance and improvements in glucose metabolism, as well as reductions in inflammation and oxidative stress, among other things.

In one of these studies, [84] ([84]) examined the relation between specific health parameters and cognitive improvements for obese, high-risk older adults. The combination of diet and exercise resulted in improvements in overall cognitive ability (composite), executive functions, and word fluency. Within the exercise groups (with and without diet) increases in VO_2_ max—which is related to the flow of oxygen to mitochondria—and increased strength—physical exercise is helpful for insulin regulation—explained 19% to 24% of the improvement in global cognitive functions. The interventions did not, however, improve inflammation, possibly because many participants remained obese, although they had lost significant amounts of weight.

Although much remains to be learned, studies that restore glucose homeostasis are associated with improvement in cognitive functions or slower age-related declines in cognition. Various aspects of mitochondrial functioning, including energy production, are integral to these improvements but are not the only source of them. At this time, it may be difficult to disentangle the relative contributions of one mechanism (e.g., mitochondrial energy production) or another (e.g., improved sensitivity to insulin) to cognitive improvements, given their interdependency. Nonetheless research in this area provides a means to test the predictions laid out in my original proposal ([47]).

## 4. Adaptive Function of Intelligence

[78] ([78]) correctly note that there are evolved mechanisms to ensure an adequate energy supply, especially to key organs such as the heart and brain. They further suggest that most individuals, during our evolutionary history, were healthy and not subject to conditions that would compromise mitochondrial energy production or other functions. This is not correct. Common threats to populations living in natural conditions, human and nonhuman, include chronic parasitic infections, nutritional and caloric deficits, and chronic social stressors (e.g., competition for mates, parenting; [45]; [115]), all of which can compromise one or several aspects of mitochondrial functioning ([46]; [56]; [66]). As noted, reductions in the severity of these threats over the 19th and 20th centuries might have contributed, at least in part, to the Flynn effect.

[113] ([113]) suggested that there was no adaptive use for intelligence, at least from an evolutionary perspective. Most generally, the large caloric costs associated with the development and maintenance of the human brain must be balanced against substantial benefits, at least in some contexts ([70]); such a costly organ could not have evolved without adaptive benefits.

There are several models that attempt to understand the adaptive importance of intelligence, most of which focus on foresight, planning, and the ability to cope with social and ecological change and novelty ([2]; [4]; [7]; [40]; [44]; [63]; [65]; [93]; [97]). The models thus primarily focus on fluid intelligence rather than *g* per se, but discussion of these models is nevertheless useful as related to Sternberg’s question.

The foci of different models of the evolution of intelligence range from the ability to anticipate and prepare for variation in seasonal changes (e.g., winter, [64]; [93]) to the protracted learning needed to become proficient in obtaining the staples of life (e.g., hunting; [65]) to competition with other people or groups of people ([2]; [40]; [44]). It is likely that some combination of these factors contributed to the evolution of fluid intelligence, although their relative importance may have changed across evolutionary time.

My attempt to integrate these different models is summarized in Figure 2. In keeping with earlier proposals ([44]; [63]) and [18]’s ([18]) original insight, the key idea is that the evolutionary function of fluid intelligence is to cope with social and ecological variability or situations that cannot be addressed by evolved or learned responses. These are situations in which automatically invoked solutions need to be inhibited and new solutions constructed. This rudimentary ability to solve novel problems is found in many species of primate and thus was almost certainly present in our hominin ancestors ([14]).

One key issue that needs to be addressed with these models is the three-fold increase in brain size over the past 4 million years of hominin evolution and the accelerated pace of enlargement over the past 2 million years, with the emergence of key *Homo* species (e.g., [3]; [79]). Although brain size and fluid intelligence are only moderately correlated ([51]; [68]) and other factors (e.g., neural density) are important ([31]), there were almost certainly evolutionary gains in fluid intelligence during hominin evolution.

One argument is that the evolutionary enlargement of hominin brain size was driven by ecological change—reductions in forested area and increases in savannah in sub-Saharan Africa—associated with climatic fluctuations before the emergence of *Homo* (e.g., [122]). One difficulty with the climate argument is that sympatric (living in same ecology) species of primate did not experience the same change in brain size ([35]). In other words, why did our australopithecine and *Homo* ancestors show marked increases in brain volume when other primate species living in the same ecologies did not?

The most straightforward answer is a within-species arms race that accelerated after the emergence of *Homo*. Species within this genus are important because they developed relatively sophisticated tools that likely reduced mortality rates and supported increased population sizes ([2]; [40])—improvement in the ability to extract resources from the ecology and reduce predatory threats is what Alexander called ‘ecological dominance’.

There is in fact consistent archeological evidence for changes in our ancestors’ ability to obtain resources from the ecology (see [42]; [43]). One result of these and related innovations, especially cooking, is an increase in the availability of calories and the energy produced by mitochondria, which would have removed an evolutionary constraint on the expansion of the energy-demanding brain ([1]). Another result is an increase in the carrying capacity of these ecologies that in turn is typically associated with population increases and expansions. As *Homo erectus* and later *Homo sapiens* expanded into new ecologies and became increasingly skilled at exploiting them, an evolutionary Rubicon was crossed:
the ecological dominance of evolving humans diminished the effects of ‘extrinsic’ forces of natural selection such that within-species competition became the principle ‘hostile force of nature’ guiding the long-term evolution of behavioral capacities, traits, and tendencies.([2])

Evidence in support of this argument is found in the pattern of human migration and the subsequent extinction of megafauna ([75], [76]), and parallels [73]’s ([73]) analysis of island biogeography. When a species migrates into an unexploited region (e.g., an island) there are few constraints on population expansion, low levels of social competition, and a rapid increase in population size, as shown by the top oval in Figure 2. As the population expands, the quantity of per capita resources necessarily declines and competition for access to these diminishing resources necessarily intensifies.

The increase in social competition is represented by the rightmost oval of Figure 2. In this circumstance, [25]’s ([25]) conceptualization of natural selection as a “struggle for existence” becomes in addition a *struggle with other human beings for control of the resources that support life and allow one to reproduce* ([44]). The eventual result of this struggle is a population crash, as was described by [74] ([74]) for many human populations in developing nations before the demographic shift and [53] ([53]) for hunter–gatherer societies. Following the crash, the per capita availability of resources is higher than it was before the crash and thus another cycle of population expansion to the carrying capacity of the ecology and later contraction begins ([38]).

The repeating cycle accelerates evolutionary selection and will favor those individuals who gain competitive advantage over others; the latter specifically refers to greater social influence and enhanced control over culturally important resources ([50]). There are many ways to achieve advantage and at least some of these are likely facilitated by fluid intelligence. As shown in Figure 2, my proposal is that fluid intelligence facilitated the creation of novel technologies, social strategies, and patterns of social organization that not only resulted in competitive advantage but further gains in these areas. We cannot of course directly study the associated social dynamics among early species of *Homo* or early modern humans, but we can look for such dynamics in the historical record.

The development of large-scale agriculture provides an apt example. Although the quality of the associated diet was often lower than could be achieved through hunting and gathering, it enabled a change in human social organization—larger groups have a competitive advantage over smaller ones—and increased the overall availability of calories that in turn supported population expansions ([21]). Critically, these additional calories could be stored as grain reserves or in livestock and these created a temping source of wealth for the taking ([57]; [117]). In areas in which steppes occupied by nomadic herders abutted fertile agricultural lands, nomadic groups and agricultural communities came into contact and oftentimes eventual conflict ([117]). The historical record shows that many nomadic groups began to raid agricultural settlements and the theft of these communities’ resources created benefits for the formation of larger agricultural communities. To counter the defensive advantage of larger communities, smaller nomadic groups had to unite to continue their raiding.

This type of cycle appears to have independently emerged in many parts of the world and was associated with advances in social organization, military strategy and technology (e.g., chariot), and eventually led to the formation of empires (see [24]; [55]; [117]; [118]). The historical record and the implied intensity of social competition (especially for men; see [50]) is supported by numerous population genetic studies showing the replacement of one human population by another in all parts of the world in which it has been studied ([130]; [131]). In other words, studies in these areas are consistent with a long-term within-species arms race. The cultural advances that emerged during the arms race must have been aided, at least in part, by individuals with a combination of strong fluid abilities and other traits (e.g., conscientiousness and planning) that support innovations in social organization and technology.

## 5. Discussion

The positive manifold and the associated covariation among cognitive and academic measures is one of the most replicated findings in the behavioral sciences ([124]). Considerable advances have been made in identifying the brain and cognitive systems that contribute to the manifold (e.g., [62]) but, as noted in all of the commentaries, much remains to be determined. The bulk of the associated theory and research has been focused on proximate cognitive processes, the broad neural networks that support these processes, and on real-world correlates, such as performance in occupational settings. These are all critical aspects of theoretical and empirical work in the field of intelligence, but there is much to be gained by expanding this work to different levels of analysis, from the dynamics of cell biology to the evolutionary function of intelligence and cognition more broadly.

My proposals here and elsewhere ([44], [47]) are first approximations of how we might more fully understand intelligence by expanding the nomological net to capture a wider range of literatures and phenomena. The first point for now is that the study of cognitive changes associated with the progression of diseases that undermine mitochondrial health provide a means to assess the plausibility of my model that in turn might provide links between these diseases and normal age-related cognitive declines and perhaps to the pattern of secular changes associated with the Flynn effect. Cognitive scientists in turn can contribute to our understanding of disease progression by designing psychometric assessments that cover the broad range of cognitive abilities, such as those identified in the CHC model, and experimental measures that might be more sensitive to individual and developmental differences in core abilities (e.g., attentional control) than are commonly used neuropsychological assessments ([36]).

The second point is that our understanding of intelligence will be enhanced by considering the challenges that different aspects of cognition helped to address during our evolutionary history. The proposal that the evolution of fluid abilities was driven by the benefits of enhanced cognitive and behavioral flexibility as related to coping with variable or novel social and ecological conditions ([44]; [63]) is consistent with recent comparative studies ([14]). The consideration of comparative work and potential evolutionary processes that drove changes in brain and cognition may be useful for placing psychometric and cognitive studies of human abilities in broader perspective.

## Figures and Tables

**Figure 1 jintelligence-08-00042-f001:**
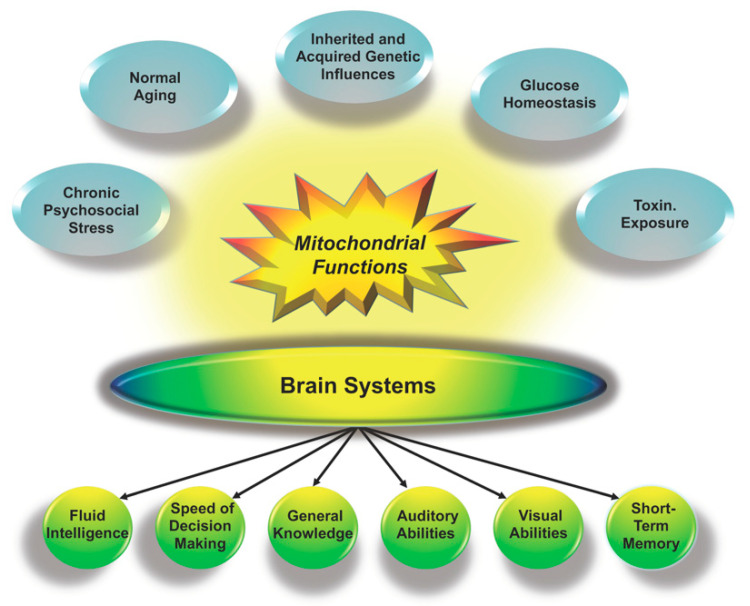
Mitochondrial health contributes to the functioning of brain systems at all levels from individual neurons to large-scale brain networks that support cognitive abilities. Mitochondrial health in turn is influenced by myriad factors, as illustrated by the surrounding ovals.

**Figure 2 jintelligence-08-00042-f002:**
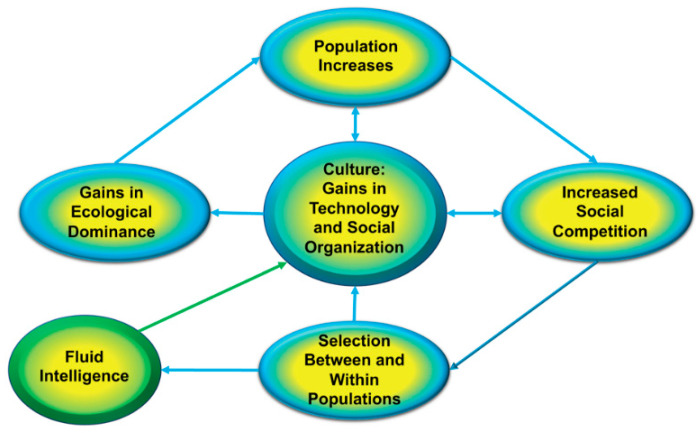
A cyclical within-species arms race is the most plausible account of the evolutionary enlargement of the human brain and enhancement in fluid intelligence.

**Table 1 jintelligence-08-00042-t001:** Core critiques and summary of key replies.

Authors	Core Critiques	Key Reply Points
Burgoyne and Engle	a. How can the hypothesis be falsified? b. What is the effect size directly related to variation in mitochondrial (mt) functioning?	a1. Strategies to falsify the hypothesis are in Table 1 of [47] ([47]). See also Section 2.2 and Section 3. a2. Unknown; some proportion of the variance associated with the *g* factor.
Debatin	a. Systems approach to neuroenergetics is more comprehensive.	a1. Agreed, but mitochondrial functioning is central to this system. This is now noted in the introduction.
Matzel et al.	a. Multiple lower-order systems influence brain and cognition. b. Variation in energy production is not sufficient to place constraints on brain and cognition. c. Not sufficient variation in mtDNA genes to create a bottleneck in energy production. d. Processes closer to (e.g., reaction time) mt should be more predictive of intelligence. e. Should not smarter people should run faster? f. Should not intelligence be more strongly correlated with mothers’ than fathers’ intelligence?	a1. Correct, but their functioning is dependent on cellular energy production. b1. Probably true for many young people in wealthy and healthy populations. b2. Probably not true with normal aging, various health conditions, nutritional deficits, parasite and toxin exposure, and myriad other stressors that are common outside of Western middle-class populations. c1. Most mitochondrial functions are dependent on nuclear not mitochondrial genes. d. The prediction is that the most complex processes will be the better predictor. e. No. Efficient energy production will only help if there is also sufficient muscle mass and mix of slow- and fast-twitch muscle fiber. f. No. Most mt genes are nuclear and inherited from both parents (Section 2.2).
Savi et al.	a. There are cyclical biological mechanisms other than mt. b. mt place a ceiling on brain functions but the ceiling might never be reached. c. This is verbal speculation. d. A dynamic, network approach to intelligence is preferable.	a. True but their operation is dependent on energy availability. b. True in many cases. However, the ceiling appears to drop with normal aging and disorders that effect mt functions or substrates and may be raised with some interventions. c. Section 3 outlines ways to test the hypothesis. d. This is not incompatible with mt; the more complex the network the more energy needed to build and maintain it.
Schubert and Hagemann	a. Many cognitive systems rely on energy consuming long-distance brain networks and remain stable or gain with aging, such as vocabulary. b. Are intervention studies of healthy adults supportive of the hypothesis? c. Should not intelligence be more strongly correlated with mothers’ than fathers’ intelligence?	a. True, but retrieving a vocabulary definition, for example, has lower prefrontal engagement than using vocabulary knowledge to solve analogies. The differences in resource demands should be differentially compromised, with normal aging; the latter more than the former. b. See Section 3. c. No. Most mt genes are nuclear and inherited from both parents (Section 2.2).
Stankov	a. Broad theories of intelligence are preferable to *g*, which is weaker than stated. b. It is important to identify diseases that directly involve mt and directly link these to cognition. c. Strong claims regarding mt and cognitive aging are premature.	a. Agreed that the study of cognition must include multiple abilities, but a substantial *g* factor remains important. b. Agreed. See Section 3. c. Agreed. This is a proposed mechanism, not a theoretical edict.
Sternberg	a. Correlational and not causal model. b. Are individual differences in intelligence related to mt for any given age? c. One’s living environments are important for cognition. d. What is the evolutionary advantage of intelligence?	a1. Agreed that much of the evidence is correlation. a2. Methods to assess a causal relation are described in Section 3. b. They could be, if people vary in factors that influence mt functioning; See Section 3. c. True, as described in Section 2.1.3. d. See Section 4.
Ujma and Kovacs	a. Genetic evidence does not strongly implicate mt functioning in intelligence. b. There is no straightforward link between complexity of cognitive tasks and mt functioning.	a1. These studies are focused on cognitive phenotypes and thus are biased toward identifying higher-level systems. a2. The bottom-up focus on mt gene-product proteins reveals a relation between mt and cognition; see Section 2.2.1. b. Correct. This is an hypothesis and some methods in Section 3 might be used to test it.

Note: mt = mitochondria.

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
