# Peer review of "Mitochondrial Functions, Cognition, and the Evolution of Intelligence: Reply to Commentaries and Moving Forward"

_jintelligence, 2020, doi:10.3390/jintelligence8040042_

Round 1
Reviewer 1 Report
This paper is a reply to commentaries on the author's previous publication in the journal. Overall, this is an interesting topic and the initial publication, the commentaries, and the reply make a nice contribution to the journal. However, I did not find this reply to be very compelling. It is longwinded, the responses to the commentaries are rather indirect, and at times in the article it is not clear what point the author is trying to make. Specific concerns:
#1
I found Section 2 to be extremely confusing. This section is supposedly about the relative importance of mitochondria but it meanders into a discussion about the meaning of g, the CHC model, and the Flynn effect. What are the main arguments in this section? Also, it wasn’t clear to me what the author’s position is on the interpretation of g; is it a general ability or is it is a composite score? The three paragraphs on page 3 are not clear at all on this point.
#2
I also found Section 4 to be unclear (and not necessary). There is a lot of speculation about the adaptive nature of intelligence and discussion of evolutionary psychology. How much of this section is really necessary? Also, it seems to me that the entire section on evolutionary psych assumes that intelligence is universal and applies to all cultures. Where is the evidence of this? It seems to me that intelligence, as currently defined and measured, is culture-dependent.
#3
Overall, the paper is far too long. A reply to commentaries should not require 14 single spaced pages.
Author Response
Comments and Suggestions for Authors
This paper is a reply to commentaries on the author's previous publication in the journal. Overall, this is an interesting topic and the initial publication, the commentaries, and the reply make a nice contribution to the journal. However, I did not find this reply to be very compelling. It is longwinded, the responses to the commentaries are rather indirect, and at times in the article it is not clear what point the author is trying to make. Specific concerns:
Overall, I have cut substantial parts of sections 2 and 4 and tried to make tighten the descriptions in other sections. I also broken section 2 into subsections to make the argument easier to follow. I also added a Table (Table 1) that summarizes the main points of each of the commentaries and the key aspects of my reply.
#1
I found Section 2 to be extremely confusing. This section is supposedly about the relative importance of mitochondria but it meanders into a discussion about the meaning of g, the CHC model, and the Flynn effect. What are the main arguments in this section? Also, it wasn’t clear to me what the author’s position is on the interpretation of g; is it a general ability or is it is a composite score? The three paragraphs on page 3 are not clear at all on this point.
As noted, I cut 3 to 4 paragraphs from this section and added subheaders to help to organize my reply.
#2
I also found Section 4 to be unclear (and not necessary). There is a lot of speculation about the adaptive nature of intelligence and discussion of evolutionary psychology. How much of this section is really necessary? Also, it seems to me that the entire section on evolutionary psych assumes that intelligence is universal and applies to all cultures. Where is the evidence of this? It seems to me that intelligence, as currently defined and measured, is culture-dependent.
I included section 4 because several of the commentaries asked about the adaptive function of intelligence. Thus, I’ve kept the basics of the section but trimmed about ½ of it. As to the cultural relevance, I can’t say. That said, I focus on the fluid abilities that is the ability to cope with novelty in a problem-solving context. It seems to me that this is relevant across cultures and appears to be present across primate species.
Burkart, Judith M., Michele N. Schubiger, and Carel P. van Schaik. 2017. The evolution of general intelligence. Behavioral and Brain Sciences 40: e195. doi:10.1017/s0140525x16000959.
#3
Overall, the paper is far too long. A reply to commentaries should not require 14 single spaced pages.
As noted, I’ve attempted to trim 2 of the main sections. The reply is still longer than would be typical for such a reply, but I thought it might be better to include more than less detail. Given the complexity of the associated issues, a brief reply is likely to raise more questions than answers.

Reviewer 2 Report
Review of
“Mitochondrial Functions, Cognition, and the Evolution of Intelligence: Reply to Commentaries and Moving Forward”
(Submitted for publication to Journal of Intelligence, 855030)
I was asked to review the reply by David Geary to the multiple commentaries on his target article Geary (2018): “Efficiency of mitochondrial functioning as the fundamental biological mechanism of general intelligence (g)”, published in Psychological Review.
In addition, a second article by Geary (2019) was considered: “Mitochondria as the linchpin of general intelligence and the link between g, health, and aging”, published in the Journal of Intelligence.
The Journal of Intelligence invited several scholars in the field to comment on these two target papers and I was invited to review Geary’s commentaries to his target articles (2018, 2019). I generally admired the broad scope of Geary’s original article(s) and also of his reply to the commentaries, all including knowledge from a large variety of fields from genetics, evolutionary biology, neuroscience and psychology up to very specialized medical research on function and development of mitochondria. I also read with large interest the commentaries and in the following I will not address all these commentaries and criticisms raised by the commentators of the target paper but I will focus only on what seem to me the most essential and critical points in Geary’s hypothesis.
In the course of this review I will argue that Geary while addressing some of the criticisms (part of them in a satisfying way) he is omitting a number of several reasonable criticisms. I will then move on to my general impression that the second part of Geary’s reply where he tries to advance his theory might be interesting in itself, but regarding the main purpose of a reply I did not really detect the connection to the initial hypothesis.
All in all – in my impression – this was less reply than rather a new target article and I generally wished that Geary took more effort to address the very reasonable points raised by the reviewers.
In the following I will only point out what in my view were the most serious criticisms of Geary’s hypothesis and whether and how they were addressed in Geary’s reply (note that this list is not exhaustive).
- Commentary by R.J. Sternberg
Sternberg in his commentary raised a most central point regarding any new theory on the biological basis of human intelligence (and of course other individual differences variables as well), namely that the differences in the suggested mechanism “are actually causal, at a meaningful level of individual or developmental differences in intelligence” (p. 1). The author has addressed this criticism (lines 140ff.) very shortly without going much into details.
The second central criticism of Sternberg is that Geary failed to present solid evidence that the mitochondrial functioning is the cause of individual differences in intelligence for a given chronological age. As correctly pointed out by Sternberg, the reasons for the ontogenetic variation in intelligence might be quite different than those for a variation within a certain age group. In intelligence research we know for example the ontogenetic development of speed of information (SIP) processing can explain the development of g with age much better than SIP can explain g-variation within an age group. Maybe I missed something but I did not find this point properly addressed in Geary’s reply.
- Commentary by Savi et al.
The most central point in Savi’s reply to me seems assumption 3 “ceilings ≠ ceiling effects”. As Savi et al. argue this is a very strict assumption. It is not clear whether the relationship between energy in the brain and intelligence is in fact a linear one or whether at some level there might be threshold (e.g. like some scholars assume for the intelligence x creativity relation).
One plausible explanation for example for the well-known Flynn effect is that the secular increase in intelligence throughout the 20th century was the availability of necessary nutrients for which there was a scarcity in the times of world war 1 and 2 (and the period between and 1 to 2 decades after); and the fact that the Flynn effect leveled off from 2000 on was also explained by the fact that from 1980, 1990s on most people are supplied with the relevant nutrients. Further improvement in nutrition might not necessarily have a further enhancing effect regarding an improvement in intelligence. Again, I did not find the point “ceilings ≠ ceiling effects” addressed in Geary’s reply.
- Commentary by Stankov
Stankov’s main criticism is about the “weakness of g” which he estimates at about 35 percent of common variance; and that in reality g is not a strong as Geary has assumed.
Here, I am with Geary and think that the estimates that he applied correspond well to the literature and his reply to Stankov addresses this criticism satisfactorily.
- Commentary by Matzel et al.
The most central point of Matzel’s commentary seems to me the reference to alternative proposals to explain g, namely van der Maas’ mutualism theory, the gene-environment multiplier theory by Dickens and Flynn, the process-overlap theory by Kovacs and Conway and the network model of intraindividual growth by Savi et al.
Again, apart from a very short attempt of explanation in lines 98-105 of Geary’s reply I would like to see more how Geary would reconcile his mitochondria hypothesis with these – partially quite influential - alternative and multifactorial explanations of g.
- Commentary by Ujma and Kovacs
A psychometric perspective is brought to the scene by Ujma and Kovacs. They point out that an important missing piece in Geary’s theory is an explanation why differences between g loadings of intelligence subtests “might depend on the relative involvement of mitochondrial function or energy”. Here, Geary argues that gf tasks are generally more difficult and therefore energy-consuming than gc tasks.
I would agree with Ujma and Kovacs’ criticism that Geary’s explanation (tasks with stronger g loadings are the result of these tasks higher energy requirement) is tautological.
While Geary elegantly embeds his hypothesis in clinical (medical), neurobiological and evolutionary theory, I see insufficient explanations how he would reconcile his hypothesis with general findings in psychometrics in intelligence research. Ujma and Kovacs plausibly ask “If mitochondrial energy is more important for certain tasks and specific ability factors that are also most strongly related to g, then it should be explained why that is the case”. This central point should be addressed more clearly in Geary’s reply.
- Debatin’s commentary
Finally, it would have been nice to see how Geary reconciles his mitochondria hypotheses with the alternative (or complementary) suggestion by Debatin for an ATP hypothesis.
Summary: I pointed out here only some very central criticisms focusing on one or two major points emphasized by most of the commentators, as mentioned my list is not exhaustive. (It could be mentioned that Geary did also not refer to all commentators). My general impression is that Geary’s reply in larger parts is more a sort of another target article, than a reply to the partially very convincing arguments from the critics. While Geary certainly proposed a very stimulating hypothesis I missed an explanation why this hypothesis should be superior to (or more fundamental than) other attempts to explain intelligence on the basis of amounts of white matter/myelin, versus grey matter, the dendrite hypothesis and several others that can be found in the literature. Why should mitochondrial function fare better to explain for example real life criteria, development with age, relations with health etc. than the other accounts that have been proposed?
Regarding the ‘moving forward’ in the title of Geary’s reply I did not see how switching the focus to other abilities like emotional intelligence (EI) bears a relationship with the mitochondria hypothesis. In the context of a ‘reply’ I found Geary’s enhancement of his hypothesis towards ‘Folk psychology’ and especially to the field of Emotional Intelligence not helpful as to me it did not become clear what role mitochondrial function could play here. Moreover, in the field of emotional intelligence there is no clear answer yet whether emotional intelligence is related to cognitive intelligence at all (see the discussion around ability vs. mixed vs. trait approaches of EI). Extending the mitochondria hypothesis to the field of Emotional Intelligence might even run into the problem of ‘who explains everything finally explains nothing’ (K. Popper).
Author Response
Comments and Suggestions for Authors
Review of
“Mitochondrial Functions, Cognition, and the Evolution of Intelligence: Reply to Commentaries and Moving Forward”
(Submitted for publication to Journal of Intelligence, 855030)
I was asked to review the reply by David Geary to the multiple commentaries on his target article Geary (2018): “Efficiency of mitochondrial functioning as the fundamental biological mechanism of general intelligence (g)”, published in Psychological Review.
In addition, a second article by Geary (2019) was considered: “Mitochondria as the linchpin of general intelligence and the link between g, health, and aging”, published in the Journal of Intelligence.
The Journal of Intelligence invited several scholars in the field to comment on these two target papers and I was invited to review Geary’s commentaries to his target articles (2018, 2019). I generally admired the broad scope of Geary’s original article(s) and also of his reply to the commentaries, all including knowledge from a large variety of fields from genetics, evolutionary biology, neuroscience and psychology up to very specialized medical research on function and development of mitochondria. I also read with large interest the commentaries and in the following I will not address all these commentaries and criticisms raised by the commentators of the target paper but I will focus only on what seem to me the most essential and critical points in Geary’s hypothesis.
In the course of this review I will argue that Geary while addressing some of the criticisms (part of them in a satisfying way) he is omitting a number of several reasonable criticisms. I will then move on to my general impression that the second part of Geary’s reply where he tries to advance his theory might be interesting in itself, but regarding the main purpose of a reply I did not really detect the connection to the initial hypothesis.
I can see where the reply was a little difficult to connect to some of the main questions raised in the commentaries. To better organize the reply, I added Table 1, which notes the main critiques (as I understand them) raised in the commentaries and my key responses. I’ve also added subheaders to section 2 which should help to organize the reply
All in all – in my impression – this was less reply than rather a new target article and I generally wished that Geary took more effort to address the very reasonable points raised by the reviewers.
As noted, I made revisions to better link the reply to main criticisms. As noted in the reply to reviewer 1, I thought it might be better to include more than less detail. Given the complexity of the associated issues, a brief reply is likely to raise more questions than answers.
In the following I will only point out what in my view were the most serious criticisms of Geary’s hypothesis and whether and how they were addressed in Geary’s reply (note that this list is not exhaustive).
- Commentary by R.J. Sternberg
Sternberg in his commentary raised a most central point regarding any new theory on the biological basis of human intelligence (and of course other individual differences variables as well), namely that the differences in the suggested mechanism “are actually causal, at a meaningful level of individual or developmental differences in intelligence” (p. 1). The author has addressed this criticism (lines 140ff.) very shortly without going much into details.
The whole point of section 3 (Empirical Studies and Testing the Hypothesis) is to discuss ways in which the hypothesis can be tested and thus provide evidence for a causal link.
The second central criticism of Sternberg is that Geary failed to present solid evidence that the mitochondrial functioning is the cause of individual differences in intelligence for a given chronological age. As correctly pointed out by Sternberg, the reasons for the ontogenetic variation in intelligence might be quite different than those for a variation within a certain age group. In intelligence research we know for example the ontogenetic development of speed of information (SIP) processing can explain the development of g with age much better than SIP can explain g-variation within an age group. Maybe I missed something but I did not find this point properly addressed in Geary’s reply.
This is correct. As noted in section 2.1.3 (Mitochondria, Environmental Conditions and the Flynn Effect), I don’t have an answer to this at this point. Nevertheless, the perspective provides a way to understand individual (at a given age) and developmental differences as related to disorders that influence mitochondrial functions (section 3) and perhaps a way to understand phenomena such as the Flynn effect and cross-national differences in mean cognitive ability.
- Commentary by Savi et al.
The most central point in Savi’s reply to me seems assumption 3 “ceilings ≠ ceiling effects”. As Savi et al. argue this is a very strict assumption. It is not clear whether the relationship between energy in the brain and intelligence is in fact a linear one or whether at some level there might be threshold (e.g. like some scholars assume for the intelligence x creativity relation).
It’s probably not linear. As I noted in Table 1 and in discussion in section 2.1.3, there are many factors that will lower the ceiling (e.g., normal aging, health deficits) and reveal individual differences in cognitive abilities that might not otherwise be found and could be related to deficits in mitochondrial functioning. I also note in section 3 that the development of mitochondrial biomarkers could provide a methods of more directly studying these relationships, whether they are linear or nonlinear. I also note in section 3.2 that there are compensatory mechanisms (e.g., mitochondrial biogenesis) that could maintain cognitive functions, despite disease progression, but then result in a precipitous (probably non-linear) decline.
In other words, Salvi et al. are correct and a direct answer to this issue is not currently available. Nevertheless, my discussion provides ways to begin to understand the complex relations between mitochondrial functions and individual and intraindividual differences (linear or not) in cognitive abilities.
One plausible explanation for example for the well-known Flynn effect is that the secular increase in intelligence throughout the 20th century was the availability of necessary nutrients for which there was a scarcity in the times of world war 1 and 2 (and the period between and 1 to 2 decades after); and the fact that the Flynn effect leveled off from 2000 on was also explained by the fact that from 1980, 1990s on most people are supplied with the relevant nutrients. Further improvement in nutrition might not necessarily have a further enhancing effect regarding an improvement in intelligence. Again, I did not find the point “ceilings ≠ ceiling effects” addressed in Geary’s reply.
I note in section 2.1.3 that nutritional and other stressors would limit the expression of genetic potential and the remediation of these stressors, as happened over the course of the demographic shift, should result in population-wide gains in cognitive abilities and health, as we saw with the Flynn effect. Once genetic potential is reached, gains would stop, as has happened.
In this section of their commentary, they ask “is there evidence that variation in the capacity for mitochondria to produce energy causes variation in intelligence.” As noted in my response to other reviewers, we don’t know yet. Nevertheless, section 3 discusses ways in which biomarkers of mitochondrial functioning might be used to address this question; and, discusses studies of disease progression that suggest that yes variation in mitochondrial functioning can influence individual differences in intelligence.
- Commentary by Stankov
Stankov’s main criticism is about the “weakness of g” which he estimates at about 35 percent of common variance; and that in reality g is not a strong as Geary has assumed.
Here, I am with Geary and think that the estimates that he applied correspond well to the literature and his reply to Stankov addresses this criticism satisfactorily.
Thank you.
- Commentary by Matzel et al.
The most central point of Matzel’s commentary seems to me the reference to alternative proposals to explain g, namely van der Maas’ mutualism theory, the gene-environment multiplier theory by Dickens and Flynn, the process-overlap theory by Kovacs and Conway and the network model of intraindividual growth by Savi et al.
Again, apart from a very short attempt of explanation in lines 98-105 of Geary’s reply I would like to see more how Geary would reconcile his mitochondria hypothesis with these – partially quite influential - alternative and multifactorial explanations of g.
I actually addressed some of these alternatives in the original article (summarized in Table 1 of that article) and refer readers to this. I list the main points of Matzel et al. and the gist of my responses in Table 1 of the reply. No doubt there are multiple factors and potential multiplier effects in the development of intelligence and cognition more broadly. I don’t, at this time, have an answer as to how these multiplier effects might interact with mitochondrial functioning.
Nevertheless, it is necessarily the case that any systems underlying these multiplier effects are dependent on adequate cellular energy supplies. Any deficits in the latter, say due to nutritional deficits, would in theory reduce the magnitude of any multiplier effects. I briefly note some of these issues in my reply in Table 1 and in section 2.1.3. Based on Reviewer 1’s request to reduce the length of the reply, this section has been trimmed and I don’t provide an extensive response to this issue, other than noting how mitochondria might be involved.
- Commentary by Ujma and Kovacs
A psychometric perspective is brought to the scene by Ujma and Kovacs. They point out that an important missing piece in Geary’s theory is an explanation why differences between g loadings of intelligence subtests “might depend on the relative involvement of mitochondrial function or energy”. Here, Geary argues that gf tasks are generally more difficult and therefore energy-consuming than gc tasks.
I would agree with Ujma and Kovacs’ criticism that Geary’s explanation (tasks with stronger g loadings are the result of these tasks higher energy requirement) is tautological.
While Geary elegantly embeds his hypothesis in clinical (medical), neurobiological and evolutionary theory, I see insufficient explanations how he would reconcile his hypothesis with general findings in psychometrics in intelligence research. Ujma and Kovacs plausibly ask “If mitochondrial energy is more important for certain tasks and specific ability factors that are also most strongly related to g, then it should be explained why that is the case”. This central point should be addressed more clearly in Geary’s reply.
I make some predictions related to this issue in section 3.2.2:
From the perspective of my model, deficits associated with disrupted glucose homeostasis (e.g., as indexed by measures of systemic inflammation) should result in a pattern of cognitive decline that varies with the energy demands of the cognitive task – these could be indexed by PET studies or the size and complexity of the supporting brain systems as identified in fMRI studies (see critique by Ujma & Kovacs, 2020) – and track the pattern of age-related cognitive declines. This would include shaper declines in fluid intelligence, measures of active learning, and attentional control and later and more shallow declines in crystallized abilities, such as vocabulary (Geary, 2018). To be sure, there are many pathological changes associated with chronic disruptions in glucose homeostasis, above and beyond declines in mitochondrial functioning, but declining mitochondrial health is an important component of the disease process (Picard & Turnbull, 2013), and the study of the pattern of cognitive declines associated with this progression provides a means to assess the plausibility of my model (Geary, 2018, 2019a).
In discussion of the Flynn effect (section 2.1.2), I predict that the relative magnitude of Flynn-effect changes, say in fluid vs. crystallized intelligence, should be the same as described in the paragraph above.
It might well be that I’m wrong, but I’d like to stick with the predictions here.
- Debatin’s commentary
Finally, it would have been nice to see how Geary reconciles his mitochondria hypotheses with the alternative (or complementary) suggestion by Debatin for an ATP hypothesis.
Right, I didn’t receive this until after I submitted the reply. Debatin makes a great point, which I address in the second paragraph of the introduction.
Summary: I pointed out here only some very central criticisms focusing on one or two major points emphasized by most of the commentators, as mentioned my list is not exhaustive. (It could be mentioned that Geary did also not refer to all commentators). My general impression is that Geary’s reply in larger parts is more a sort of another target article, than a reply to the partially very convincing arguments from the critics. While Geary certainly proposed a very stimulating hypothesis I missed an explanation why this hypothesis should be superior to (or more fundamental than) other attempts to explain intelligence on the basis of amounts of white matter/myelin, versus grey matter, the dendrite hypothesis and several others that can be found in the literature. Why should mitochondrial function fare better to explain for example real life criteria, development with age, relations with health etc. than the other accounts that have been proposed?
I’ve added Table 1 to highlight my replies to major comments and provided subheadings in section 2 to make these points more obvious (they were muddled in my initial reply).
True, I did add more than might be necessary and went beyond a typical reply. As I noted in my reply to Reviewer 1 and in my original article, any relation between mitochondrial functions and cognition is complex and nuanced. A shorter reply that did not capture some of this nuance and clearly explain methods that can be used to test the mitochondria hypothesis would create more questions than answers.
In other words, I think it’s better to over explain, given the complexity of the issue, than to provide shorter responses that might result in more confusion than clarity.
Regarding the ‘moving forward’ in the title of Geary’s reply I did not see how switching the focus to other abilities like emotional intelligence (EI) bears a relationship with the mitochondria hypothesis. In the context of a ‘reply’ I found Geary’s enhancement of his hypothesis towards ‘Folk psychology’ and especially to the field of Emotional Intelligence not helpful as to me it did not become clear what role mitochondrial function could play here. Moreover, in the field of emotional intelligence there is no clear answer yet whether emotional intelligence is related to cognitive intelligence at all (see the discussion around ability vs. mixed vs. trait approaches of EI). Extending the mitochondria hypothesis to the field of Emotional Intelligence might even run into the problem of ‘who explains everything finally explains nothing’ (K. Popper).
I agree that this is a bit of a stretch and not necessary – the goal wasn’t to extend it to mitochondria but to say that an evolutionary perspective might provide a broader net when thinking about the pattern of cognitive abilities. In any case, it’s been deleted from the reply.

Round 2
Reviewer 1 Report
The author has done an excellent job responding to my previous concerns and revising the manuscript accordingly. This is a really nice reply to the commentaries and I think will make a major contribution to field. Well done!
Reviewer 2 Report
In his revision of the paper the author did a very god job to address all my concerns regarding the first version of the paper. The additions as well as the ‘pruning’ in some passages make the paper/reply now much more succinct and I recommend publication.